# Bran-Enriched Milled Durum Wheat Fractions Obtained Using Innovative Micronization and Air-Classification Pilot Plants

**DOI:** 10.3390/foods10081796

**Published:** 2021-08-03

**Authors:** Alessandro Cammerata, Francesco Sestili, Barbara Laddomada, Gabriella Aureli

**Affiliations:** 1Council for Agricultural Research and Economics, Research Centre for Engineering and Agro-Food Processing, Via Manziana 30, 00189 Rome, Italy; alessandro.cammerata@crea.gov.it; 2Department of Agriculture and Forest Sciences (DAFNE), University of Tuscia, Via San Camillo de Lellis snc, 01100 Viterbo, Italy; francescosestili@unitus.it; 3Institute of Sciences of Food Production (ISPA), National Research Council (CNR), Via Monteroni, 73100 Lecce, Italy

**Keywords:** durum wheat, milling fractions, air-classification plant, micronization plant

## Abstract

Dietary guidelines recommend the consumption of unprocessed, or minimally processed, wheat foods because they are richer in health-promoting components (i.e., minerals, vitamins, lignans, phytoestrogens, and phenolic compounds) compared to traditionally refined products. The design and implementation of technological solutions applied to the milling process are becoming a key requirement to obtain less refined mill products characterized by healthier nutritional profiles. This study presents the development of an upgraded micronization plant and of a modified air-classification plant to produce several novel types of durum wheat milling fractions, each enriched in bran particles of different sizes (from 425 µm > Ø to Ø < 180 µm) and percentage ratios. A preliminary quality assessment of the milling fractions was carried out by measuring yield percentages and ash content, the latter being related to detect the presence of bran particles. A wide array of milling fractions with different original particle size compositions was provided through the study of the process. Results indicate the ability of the novel pilot plants to produce several types of less refined milling fractions of potential interest for manufacturing durum wheat end-products beneficial for human health.

## 1. Introduction

Durum wheat (*Triticum turgidum* L. ssp. *durum*) is the most important cereal staple food in Mediterranean climates and regions. Traditionally, it is the primary raw material used in the production of pasta, couscous, bulgur, and different types of leavened and unleavened breads [1]. Overall, these products provide a significant portion of calories and proteins to human diets; also, they are an important source of bioactive compounds that may contribute to a healthy diet [2]. Among others, the most common bioactive compounds of wheat include dietary fiber, vitamins, micronutrients, and phytochemicals, which are mainly located in the outer layers of the kernel, typically in the bran, aleurone, and germ tissues [3]. A high number of in vitro, in vivo, and epidemiological studies have shown the significant health-related benefits associated with the consumption of bran-rich or whole-wheat foods [4], together with a decreased risk of non-communicable diseases, such as type 2 diabetes mellitus, cardiovascular disorders, and colorectal cancer [5]. Despite the known positive effects, the consumption of whole wheat food products is still limited in many countries. Such a hindrance may depend on consumer knowledge and behaviors [6], but also on the negative impact of wheat bran on the sensorial quality of end products [7]. Several studies have evaluated the effect of bran particle size, bran pre-treatment, and cooking method on the sensorial quality of final products [8]. The interfering effect of bran on both protein hydration and dilution of gluten network in the dough is due to the occurrence of hydroxyl groups in the bran structure reacting with water through hydrogen bonds, resulting in an increase in water absorption. In fact, arabinoxylans reduce the amount of water available for the gluten network, thus affecting dough stability and development time [9]. New technological solutions in wheat milling processing have many advantages as compared to traditional milling, which, by removing germ and bran, cause the loss of several beneficial compounds that are located in the bran fraction. Industrial pre-treatments, such as debranning, micronization, and air-classification processes, can reduce the negative effects of wheat bran [10,11,12,13]. Debranning is a dry separation technology based on consecutive abrasions of cereal kernels; the progressive bran removal through the detachment of the outer, intermediate, and inner layers of pericarp leads to different byproduct classes, which can be removed by using pressurized air flowing through the screens and outlets of the debranner [7].

The reduction in particle size obtained through micronization can be followed by an air-classifier treatment in order to obtain two or more fractions that are collected separately. Micronization consists of a milling process able to reduce the starting matrix such as cereal grains in a fine particle product through the use of different technologies (e.g., hammer mill, knife mill). The range of the particle size of the milled product depends on the sieve diameter employed [11]. The general functioning criteria of the air-classification system is based on the pneumatic transportation of the milled particles inside the plant, which operates in depression, thus promoting a pneumatic flow along circular orbits. Inside the orbits, a series of ascending airflows was controlled by setting the airflow inlet valve to different opening conditions. Due to the combined action of different forces, the heavier particles fall in the first “G” housing (heavier gross particles), whereas the lighter particles fail to be collected in the first step but are gathered in the second “F” housing (fine particles). Bran fractions derived from both debranning and micronization were used to enrich semolina, obtaining pasta products with enhanced health properties and minimal impacts on sensory quality [14]. The above technologies are also useful to better manage the natural or chemical contaminants (e.g., mycotoxins, heavy metals, and pesticides) that are typically concentrated in the outermost layers of the wheat kernel [15,16,17]. 

The aim of this study was to further improve both the micronization and air-classification of plants by introducing ad hoc modifications into the process in order to obtain a larger choice of less-refined milling fractions. More in detail, the micronizer pilot plant improvement was carried out through the addition of a grinding chamber, equipped with a hammer crusher impeller and a decanting collector, whereas the air-classifier pilot plant was added with both a programmable logic controller (PLC) for the pneumatic flow management and a chamber specially designed with a decreasing section. These latest improvements of pilot plants already supplied in our laboratories allowed us to produce innovative mixtures of milling products. Particular focus was put on studying the process to develop diverse milling fractions, each characterized by a peculiar content and composition in bran particles that could be used to make durum wheat end-products characterized by a higher bran content and healthier nutritional content. 

## 2. Materials and Methods

### 2.1. Durum Wheat Cultivars

Three Italian durum wheat cultivars, namely Saragolla, Maestà, and Iride, were grown under conventional farming in the experimental field of CREA, Foggia (Italy) in the 2018–2019 growing season. Single grain samples were evaluated for test weight, moisture, protein content, gluten content, and yellow color through near-infrared analysis in transmission mode (NIT) by using Infratec™ mod 1241 (FOSS, Hillerød, Denmark) and the results were expressed as mean value of replicates (n = 10). Figure 1 shows the flow chart of the experimental plan.

### 2.2. Micronization of Grain Samples and Air Classification of Milling Fractions

Durum wheat grain samples (11 kg for each cultivar) were micronized using a micronizer pilot plan (mod. 32300, KMXi-300-7,5; Separ Microsystem S.a.s, Brescia, Italy). The micronization step did not require preventive conditioning of the grains. An improvement in the micronizing pilot plant was achieved by adding a hammer crusher impeller with a reduced cross-section inside the grinding chamber compared to the normal type. This latter was equipped with a sieving grid (Ø = 0.7 mm) suitable for obtaining a more homogeneous product. In addition, the grinding chamber was connected to a decanting chamber suitable for collecting the ground product. Moreover, with the aim to better detach the milled product from the plant walls, an electric vibrator was placed on the top side of the same chamber (Figure 2). 

Afterwards, 10.5 kg of micronized sample was submitted to an air-classifier pilot plant (turbo-separator unit, model SX-LAB; Separ Microsystem S.a.s, Brescia, Italy) suitable for particle sizes up to a set limit (Ø ≤ 1.5 mm). 

The plant was improved through the insertion of a programmable logic control (PLC) able to manage both the frequency and the action time of the airflow inside the separation chamber. Additionally, the chamber was further improved by the modification of the internal orbits that had been made according to a progressive decrease in the internal section. The latter modification allowed us to enhance the separation between the fine fractions and the coarse ones (Figure 3). 

Micronized aliquots of 1.5 kg were air-classified for one cycle at a time by setting the airflow inlet valve to 220, 230, 240, 250, 260, 270, or 280. At the end of each cycle the fractions of type G and F were collected and submitted for analysis.

### 2.3. Quality Analysis

Mean yield percentages of samples were evaluated based on the weight percentage of each sample in relation to the starting weight of the same sample. Ash content was also determined by following the official method [18]. The particle sizes of all fractions were measured by using certified test sieves (Giuliani Tecnologie S.r.l., Turin, Italy). 

### 2.4. Statistical Analysis

The analysis of variance was performed on transformed data applying ANOVA (post hoc: Tukey test and Bonferroni correction) using the statistical software PAST 2.12 [19]. The data transformation (log or root square) was needed to achieve a normal distribution of the same data suitable for statistical analysis. The Bonferroni correction was used to counteract the incorrect rejection of a null hypothesis. 

## 3. Results

The qualitative characterization of Saragolla, Maestà, and Iride grains are summarized in Table 1. Test weights among the three cultivars ranged from 81.0 to 84.4 kg/hL, namely in the first class group according to the official standards [20], revealing the absence of significant quantities of shrivelled kernels. The moisture content of the grains varied from 10.2% to 10.9%, below the maximum limit of 14.0%. 

Except for yellow color, which was low for each of the three cultivars, protein and gluten percentages were satisfactory in all samples varying from 12.6 to 15.6, and from 8.9 to 11.0, respectively, suggesting good potential nutritional and technological quality of the cultivars.

Milling yield percentages were determined to test the efficiency of the milling process. The mean recovery of micronized samples was equal to 98.8%, showing a negligible loss of starting sample.

As expected, the air-classified fractions showed marked differences between the F and G fractions (Figure 4). In detail, the F fraction showed yield values with an ascending trend from 23% (registered for setting 220) to 93% (for setting 280). Conversely, a descending trend was observed for the G fractions, varying from 77% (setting 220) to 6% (setting 280). Yield recovery of the F fractions clearly exceeded 60% only for setting conditions ranging from 250 to 280. In the G fractions, this value exceeded only at 220 and 230 settings. 

We did not observe significant differences (*p* > 0.05) among the three cultivars at any of the operating airflow conditions, with the exception of F240, G240, and G260 (Figure 4). Results obtained for each cultivar are presented in Appendix A.

Further investigations are in progress to test the reliability of the process at those conditions. 

The F and G type fractions obtained through the micronization and air-classification of whole durum grains contained a different percentage distribution of bran particles, due to the applied opening rate of the inlet valve. Ash content of F fractions was higher compared to that of G fractions, due to the higher content of bran particles in F fractions, while the G were richer in semolina (Figure 5). Considering the ash percentages across the F fractions, a descending trend was observed, with F220 (3.16%) and F230 (2.74%) showing the highest levels. Conversely, no significant differences (*p* > 0.05) were pointed out across the G fractions, observing a range from 1.67% (G220) to 1.42% (G280). Ash percentage of the air-classified fractions for each cultivar showed a trend that was similar to that of cultivar mean values with no significant differences (*p* > 0.05) from G220 to G280 (Appendix A).

Results concerning the particle size distributions (Ø > 425 µm, 425 µm > Ø > 180 µm, Ø < 180 µm) within the air-classified fractions are shown in Figure 6. As expected, heavier particles (Ø > 425 µm) were found mainly in the G fractions, because these contained more semolina than bran residues as compared to the F fractions. The percentage of the heavier particles in G fractions ranged from 11.4% (G220) to 53.4% (G280). Conversely, heavy particles were scarcely present in the F millings, varying from 4.3% (F250) to 19.9% (F220). Intermediate (425 µm > Ø > 180 µm) and fine (Ø < 180 µm) particles were predominant in all F fractions. 

Together, the intermediate and fine particles were prevalent in all G fractions, with the exception of G280 where their amount was 46.6% of the total, which could be explained by the less open inlet valve and the consequent different dynamics of the incoming airflow. Notably, a low percentage of finer particles was detected in F220 and F230 (18.8% and 38.3%, respectively). Detailed data on each cultivar are presented in Appendix A, showing a marked trend towards no significant differences (*p* > 0.05) of percentage values from G250 to G280 fractions of all particle sizes assayed. The statistical analysis highlighted the influence of the two sources of variation (cultivar and setting valve point) related to the afore mentioned three particle sizes assayed (heavier, intermediate, and finer). The results showed that each source of variation had a significant (*p* < 0.05) influence on heavier size particles (Ø > 425 µm) in the fractions F, whereas their interaction was significant (*p* < 0.05) in the case of finer size particles both in F and in G. In the case of the fractions G, only the setting valve point factor was significant (*p* < 0.05) for heavier size particles (Ø > 425 µm).

## 4. Discussion

The measure of yield percentages allowed us to evaluate the air-classified fractions from the point of view of milling quality, this latter intended as an important aspect for the choice of more suitable fractions also for large scale production. 

The ash content is an important quality parameter influenced both by genetic and environmental factors and associated with bran content [21]. In this study, the assay of the ash percentage proved useful to better characterize each air-classified fraction in any case containing both bran and semolina particles in different ratios. 

The evaluation of particle size distribution within each fraction type (F and G) revealed the key role played by the involved physical factors (i.e., weight, air-pressure inside the circuits, turbulence, etc.). The resultant force obtained by the air-classification process, intended as a whole system, determined the distribution of the particles in F or G collectors. In any case, all air-classified millings were less refined compared to traditional semolina and were characterized by a specific percentage of bran residues. In studying the processing plants performance, the use of more than one cultivar with different grain characteristics was considered to test if they could introduce possible “critical” issues influencing the reliability of the system. Although the results need to be confirmed on a higher number of cultivars grown in different years and environments, the preliminary data pointed out that the F220, F250, and G240 fractions showed significant differences (*p* < 0.05) among the cultivars. Nevertheless, the use of the innovative plants allowed the production of several different types of air-classified fractions, allowing us to choose the more suitable for making less refined and more attractive end-products with innovative properties from among them due to the different combinations of heavier, intermediate and finer bran particles. Previous studies already highlighted that the addition of durum wheat bran with a particle size range of 150–500 µm, at levels of 10%, 20%, and 30%, had a negative impact on pasta sensory and technological properties [22]. However, through the addition of 30% of the same size range of bran particles, a comparable texture to that of commercial whole milled wheat pasta was achieved. Conversely, the addition of 10% had similar sensory texture scores to regular durum wheat pasta not supplemented with bran. The use of milling fractions enriched with more fine bran particles is expected to be more suitable for the release and bioavailability of bioactive compounds, due to both the major exposition of bran particle surfaces and aleurone cells to disruption, resulting in a better release of the intra cellular contents [23]. Therefore, the different air-classified fractions here described offer a suitable tool to produce innovative food-products characterized by new technological properties and improved nutritional value that could make these products more attractive to consumers. The inclusion of bran particles in wheat millings leads to technological disadvantages and worsens end-products quality compared to refined semolina or flour-based processes and products, causing, among other effects, a decrease in bread loaf volume, textural changes, and color changes [24]. These effects are more severe, especially at 20% incorporated bran, reducing pasta quality; yet, a reduced impact occurs at the same percentage of incorporation using finer bran [21]. In general, bran supplementation also has a positive effect due to the increase in polyphenols and phytosterols in end-products. These parameters are not influenced by bran particle size above 10% incorporation, except for phenolic acids, which increase at a higher rate of finer bran particles [21]. Therefore, the range size between 180 µm and 425 µm, which is included in the fractions here assayed, has a key role due to the presence of a part of the fine bran fractions (<180 µm, 180 µm, and 250 µm) suitable for better nutritional quality of pasta due to the presence of health-promoting compounds (from the outer kernel layers) compared to traditional pasta produced with refined semolina. The influence of the bran particle size on the water absorption of cooked pasta has been deeply studied. Finer bran particles produce a lower and significant degree of water absorption compared to particles with a larger diameter, regardless of the derivation of the particles (e.g., bran or middlings) [25]. On the other hand, the positive effects of medium coarse and high coarse particle sizes of semolina on the end product (pasta) quality has already been described [26]. Therefore, the particle size composition of less-refined fractions constitutes a very important aspect to be considered, especially in relation to technological aspects and the intended use of the end-products.

With regard to the detailed results concerning the single cultivars, it should be underlined that the choice to use them was aimed only at increasing the variability of samples tested. In any case the results are intended as a study on the cultivar behavior in the process, which would require a different experimental design (greater number of cultivars grown under different and controlled environmental conditions over multiple seasons). Indeed, our study was aimed at assessing the application of new technological solutions to current micronization and air-classification plants in order to improve the quality of the overall milling performances.

## 5. Conclusions

The improvement of the micronization and air-classified plants yielded a wide range of less-refined milling fractions that could be suitable to make durum wheat end-products characterized by enhanced health-promoting qualities. The technological modifications of the pilot plants already in use in our laboratories regarding both the micronizer (through the addition of a grinding chamber including a hammer crusher impeller and a decanting collector) and the air-classifier (through the inclusion of PLC and a suitable chamber) allowed us to deepen some quality aspects linked to the new mixtures that were developed. Indeed, milling fractions (F and G) have been characterized by new particle size content and composition as compared to those obtained before the update of milling plants. A preliminary assessment of the quality of the milling fractions was based on yield percentages, ash content, and particle size composition. The results revealed that the obtained air-classified fractions offer diverse choices to make a good compromise between high yield percentage, particle size composition, and the health benefits associated with a higher content of bran (fiber, antioxidants, minerals, etc.). Further investigations on technological and qualitative features of the air classified fractions are ongoing and will be the subject of upcoming papers.

## Figures and Tables

**Figure 1 foods-10-01796-f001:**
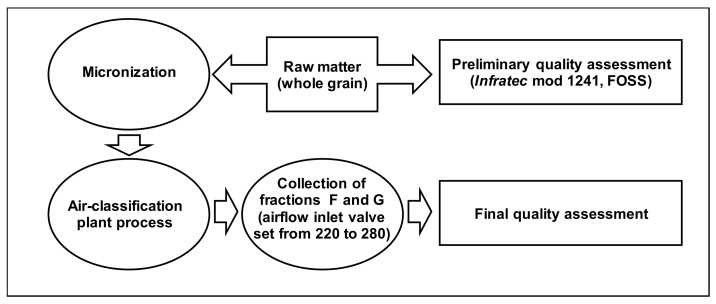
Flow chart of the experimental plan.

**Figure 2 foods-10-01796-f002:**
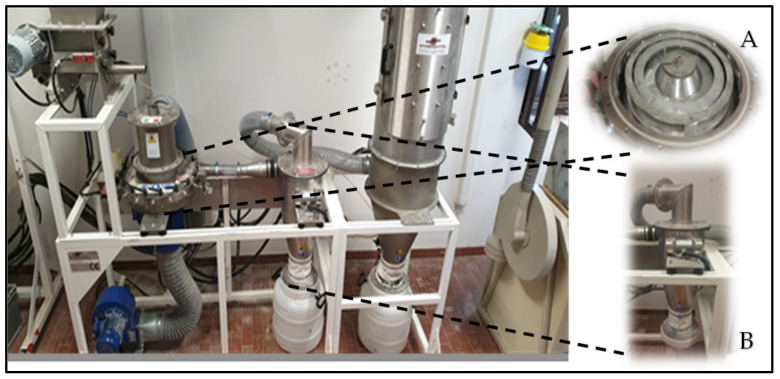
Micronizer pilot plant improved through the addition of a grinding chamber including a hammer crusher impeller (**A**) and a decanting collector (**B**).

**Figure 3 foods-10-01796-f003:**
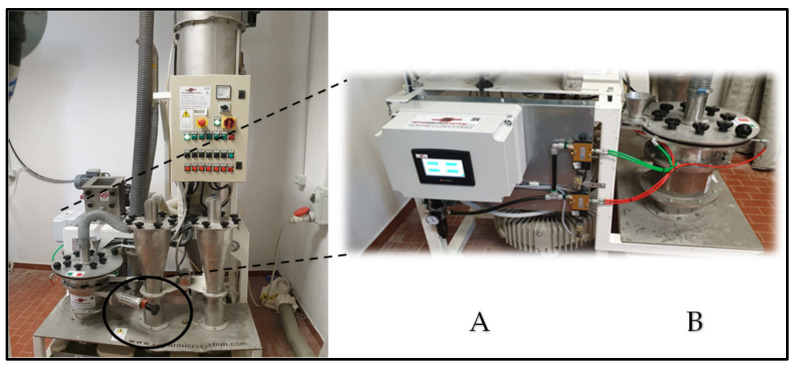
Air-classifier pilot plant augmented through the addition of a programmable logic controller (PLC) for the pneumatic flow management (**A**) and a chamber designed with a decreasing section (**B**). The circle indicates the setting airflow inlet valve. The F and G housings are placed below the steel plan located under the inlet valve.

**Figure 4 foods-10-01796-f004:**
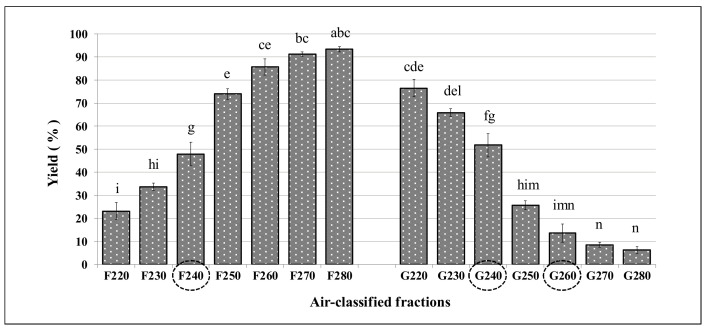
Mean yield percentage (%) of the air-classified fractions of samples belonging to the cultivars Saragolla, Maestà, and Iride. Different letters indicate statistically significant difference (*p* < 0.05, n = 3). Circles highlight the fractions with significant differences (*p* < 0.05) among cultivars (n = 2).

**Figure 5 foods-10-01796-f005:**
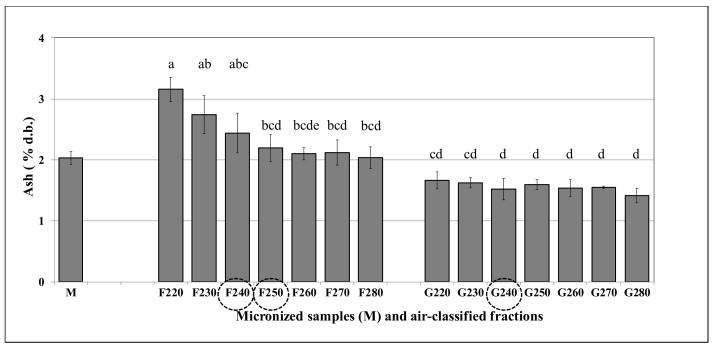
Mean ash percentages (% dry basis) of three durum cultivars: micronized wholemeals (M); air-classified fractions (F and G types). Different letters indicate statistically significant difference (*p* < 0.05); circles highlight the fractions with significant differences (*p* < 0.05) among cultivars (n = 2).

**Figure 6 foods-10-01796-f006:**
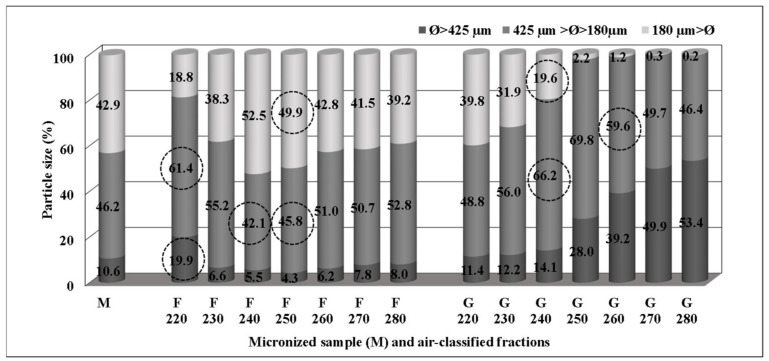
Mean percentages of particle size in micronized samples (M) and air-classified fractions (F and G types) of the mean of three cultivars. Circles highlight the fractions with significant differences (*p* < 0.05) among cultivars (n = 2).

**Table 1 foods-10-01796-t001:** Durum wheat cultivars, origin (region of Italy), and proximate quality parameters measured by NIT. Mean values of replicated analyses (n = 10).

Cultivar	Origin	Test Weight (kg/hL)	Moisture (%)	Protein Content (% db)	Gluten Content (% db)	Yellow Color
Saragolla	Puglia	81.6	10.9	12.9	8.9	13.9
Maestà	Puglia	81.0	10.2	15.6	11.0	15.4
Iride	Puglia	84.4	10.7	12.6	9.1	14.2

## Data Availability

The data are contained within the article.

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
