# Peer review of "Bran-Enriched Milled Durum Wheat Fractions Obtained Using Innovative Micronization and Air-Classification Pilot Plants"

_foods, 2021, doi:10.3390/foods10081796_

Round 1
Reviewer 1 Report
The paper ‘Bran-enriched milled durum wheat fractions obtained through the use of innovative micronization and air-classification plants’ is an attempt to verify dietary recommendations to consume unprocessed food as the food containing more healthy ingredients. The trend is very popular, the research in that area is more than welcome and can be seriously interesting for a potential reader. The manuscript is nicely written in an understandable way.
COMMENTS:
Discussion and conclusions can’t be justified and verified based on presented results. Authors used three different cultivars, but in the paper only means for all three ones are shown.
Authors should enrich the paper by presentation and discusion of results for each cultivar separately. Otherwise several unnecessary questions can appear decreasing value of the paper like in Fig 6. for G260, values for cultivars are significantly different for second category of particle size, but no differences were observed for other two categories. It seems that significant differences should be observed for at least one more category for G260.
The final recommendations is to introduce separate results for each cultivar in the entire paper and introduce the discussion for each cultivar with adequate explanation of observed differences and variations. Please adjust the Conclusions section as well.
The subject is in the circle of interest, so it is important for the reader and scientific community to understand not only the global behaviour of the phenomena, but the behaviour for individual cultivar as well.
Additional tables with results (with statistical analysis – significance, standard deviations) corresponding to each cultivar can be added in the main manusript or as an appendix.
Unfortunately, without the additional data, the paper can not be recommend for publication.
Author Response
Reviewer #1
The paper ‘Bran-enriched milled durum wheat fractions obtained through the use of innovative micronization and air-classification plants’ is an attempt to verify dietary recommendations to consume unprocessed food as the food containing more healthy ingredients. The trend is very popular, the research in that area is more than welcome and can be seriously interesting for a potential reader. The manuscript is nicely written in an understandable way.
COMMENTS:
Discussion and conclusions can’t be justified and verified based on presented results.
We have modified the discussion and conclusions. Please, find the tracked changes along the two paragraphs.
Authors used three different cultivars, but in the paper only means for all three ones are shown. Authors should enrich the paper by presentation and discussion of results for each cultivar separately.
Please, find the requested data as supplementary materials (Appendix A, B and C). Indeed, the paper is a short communication: our aim was to test the ability of the upgraded pilot plants to produce a large variety of durum wheat milling fractions using as sample a discrete number of durum wheat grains. The intention was not to test cultivar behaviour in the process, which should need more cultivars, field trials carried out under controlled environmental conditions and across different environments, over the years.
Otherwise several unnecessary questions can appear decreasing value of the paper like in Fig 6. for G260, values for cultivars are significantly different for second category of particle size, but no differences were observed for other two categories. It seems that significant differences should be observed for at least one more category for G260.
We agree with your observation, but we reported all significant differences.
The final recommendations is to introduce separate results for each cultivar in the entire paper and introduce the discussion for each cultivar with adequate explanation of observed differences and variations. Please adjust the Conclusions section as well.
The subject is in the circle of interest, so it is important for the reader and scientific community to understand not only the global behaviour of the phenomena, but the behaviour for individual cultivar as well. Additional tables with results (with statistical analysis – significance, standard deviations) corresponding to each cultivar can be added in the main manusript or as an appendix.
Unfortunately, without the additional data, the paper can not be recommend for publication.
As suggested, we have added three tables as Appendix A, B and C and we presented and commented the data concerning the cultivars along the Results and Discussion sessions, though our study was not focused on cultivar behavior requiring a greater number of cultivars and growing seasons.
Reviewer 2 Report
Manuscript foods-1283703 has undoubtedly been well written. The authors provide an Introduction that places the reader in the need for the research. The methodologies are clear and the results are discussed in an appropriate manner. However, I consider that the research lacks novelty, since the same authors have demonstrated the advantages of their proposal in other papers such as 10.1002/cche.10458 and 10.1177/1082013217745199, among others. In those papers, published by the same research group, the conclusions made in this new manuscript are already presented. So I consider that due to the high impact factor of Foods, this research does not have enough novelty to be published in this journal. This does not mean that it is a bad work, since I can see that the results are part of a large funded project, which is commented in some of the publications of the research team.
This research does not have enough novelty
Author Response
Reviewer #2
Manuscript foods-1283703 has undoubtedly been well written. The authors provide an Introduction that places the reader in the need for the research. The methodologies are clear and the results are discussed in an appropriate manner. However, I consider that the research lacks novelty, since the same authors have demonstrated the advantages of their proposal in other papers such as 10.1002/cche.10458 and 10.1177/1082013217745199, among others. In those papers, published by the same research group, the conclusions made in this new manuscript are already presented. So I consider that due to the high impact factor of Foods, this research does not have enough novelty to be published in this journal. This does not mean that it is a bad work, since I can see that the results are part of a large funded project, which is commented in some of the publications of the research team.
This research does not have enough novelty.
We apologize for not clarifying nor stressing the novelty of our work as compared with the previous published papers. The present study modified the micronizer and air-classifier plants cited in previous works of our group. The novelty is stressed along the manuscript through the addition of comments and further literature. Therefore, the micronizer and air-classifier plants, already in use in our laboratories have been subjected to a major improvement by the addition of original and ad hoc designed modifications applied to the durum wheat for the first time in this study. Moreover, the technological improvement showed in the present short communication is now undergoing to patent application.
Reviewer 3 Report
Cammerata et al. it their manuscript 'Bran-enriched milled durum wheat fractions obtained through the use of innovative micronization and air-classification plants' investigated how the pilot plant equipment can be used to produce fractions of flour differing in particle size and ash content. The work is important beginning on the hot topic of the cereal micronization and fractionation. On the other hand, it is just a good start, and the results shown here do not have a meaning. In my opinion, it needs to be seriously extended with addition data such as the repeatability of milling process, as well as the safety and functionality of product. The abstract and conclusion are not based on the results shown here. The introduction lacks the background about micronization of cereals, what was done in recent years. The materials and methods are not adequately described. The discussion is unrelated to this work. The more detail comments re given bellow.
I suggest changing the title (but depending on the change of manuscript content) e.g. Bran-enriched milled durum wheat fractions obtained through the use of innovative micronization and air-classification pilot plant
Abstract
line 22: bioactive compounds are not shown in the manuscript
Lines 22-24: How the reliability was demonstrated? How it is beneficial for human health?
It would be interesting to read what actual particle sizes and ash content in different fractions were achieved.
Line 57: The introduction on micronization is missing. What types of mills are usually used to micronize cereals? What is the range of obtained particles sizes? Please give an overview of what was done in recent years and how you work fills the gap in the existing knowledge.
Line 81-83; 88-89; 118-121 Please indicate the number of replicates done.
L 80: What were the main process parameters e.g. time, cycles, etc.
Line 112 and Fig .1. it is not clear what is F fraction and what is G fraction – how they were obtained, what was the setting difference.
Lines 219-251. The discussion in completely not related to this work.
L 261: The conclusion is not founded on the results shown in this study.
Lines 261-264. Some of those results or other important data concerning the safety of products (e.g. mycotoxins, heavy metals, pesticides residue), functionality (e.g. water absorption, swelling, dough rheology) or the bioactive content should be shown in the current paper, not future one. Microscopy pictures would be also useful.
Author Response
Reviewer #3
Cammerata et al. it their manuscript 'Bran-enriched milled durum wheat fractions obtained through the use of innovative micronization and air-classification plants' investigated how the pilot plant equipment can be used to produce fractions of flour differing in particle size and ash content. The work is important beginning on the hot topic of the cereal micronization and fractionation. On the other hand, it is just a good start, and the results shown here do not have a meaning. In my opinion, it needs to be seriously extended with addition data such as the repeatability of milling process, as well as the safety and functionality of product. The abstract and conclusion are not based on the results shown here. The introduction lacks the background about micronization of cereals, what was done in recent years. The materials and methods are not adequately described. The discussion is unrelated to this work. The more detail comments re given bellow.
I suggest changing the title (but depending on the change of manuscript content) e.g. Bran-enriched milled durum wheat fractions obtained through the use of innovative micronization and air-classification pilot plant
Thank you for the right suggestion. The title has been changed.
Abstract
line 22: bioactive compounds are not shown in the manuscript
The phrase has been corrected.
Lines 22-24: How the reliability was demonstrated? How it is beneficial for human health?
It would be interesting to read what actual particle sizes and ash content in different fractions were achieved.
The abstract has been improved.
Line 57: The introduction on micronization is missing. What types of mills are usually used to micronize cereals? What is the range of obtained particles sizes? Please give an overview of what was done in recent years and how you work fills the gap in the existing knowledge.
More information was added on the micronization process and more literature references have been cited.
Line 81-83; 88-89; 118-121 Please indicate the number of replicates done.
The number of replicates has been specified.
L 80: What were the main process parameters e.g. time, cycles, etc.
The question is not related to the sentence reported in L. 80 (where field trials are described). Please, help us to understand the question, providing the correct line number.
Line 112 and Fig .1. it is not clear what is F fraction and what is G fraction – how they were obtained, what was the setting difference.
The two fractions are described in the introduction and included in the description of the general functional criteria (methods).
Lines 219-251. The discussion in completely not related to this work.
The discussion has been revised.
L 261: The conclusion is not founded on the results shown in this study.
Lines 261-264. Some of those results or other important data concerning the safety of products (e.g. mycotoxins, heavy metals, pesticides residue), functionality (e.g. water absorption, swelling, dough rheology) or the bioactive content should be shown in the current paper, not future one. Microscopy pictures would be also useful.
This paper is a short communication aimed to describe the new plant and to evaluate the process.
However, mycotoxins and heavy metals have already studied in our previous paper that has been cited in the introduction. Other aspects regarding the milling fractions obtained in this work will be the subject of upcoming papers. In fact, our study was aimed at assessing the application of new technological solutions to current micronization and air classifications plants in order to upgrade the overall milling performances.
Round 2
Reviewer 1 Report
The revised paper ‘Bran-enriched milled durum wheat fractions obtained through the use of innovative micronization and air-classification pilot plants’ is an attempt to verify dietary recommendations to consume unprocessed food as the food containing more healthy ingredients. The scope of the paper is interesting.
The novelty of the results having in mind previous papers is not so high. Authors should provide additional experiments based on investigations on technological and qualitative features.
The comments regarding statistics and selected results are still valid.
Some minor comments like:
line 136 – The explanation what is the meaning of the term ‘transformed data’ is needed
line 137 – the explanation of applying Bonferroni correction is needed
Additional comment regarding formatting of the paper:
The paper is delivered in almost unreadable way. The corrected parts should be marked by different colour, not by following changes mode.
Author Response
The revised paper ‘Bran-enriched milled durum wheat fractions obtained through the use of innovative micronization and air-classification pilot plants’ is an attempt to verify dietary recommendations to consume unprocessed food as the food containing more healthy ingredients. The scope of the paper is interesting.
The novelty of the results having in mind previous papers is not so high. Authors should provide additional experiments based on investigations on technological and qualitative features.
This paper proposed as short communication has to be intended as a preliminary and exclusively in depth study of the updated milling process of grains here applied to durum wheat. The improvements of pilot plants, already supplied in our laboratories, allowed us to assay innovative and poor refined mixture as compared to those obtained before the updating design. The main challenge was to assess the reliability of the new designed plants through the use of a sufficient wide choice of different durum wheat samples.
The comments regarding statistics and selected results are still valid.
Some minor comments like:
line 136 – The explanation what is the meaning of the term ‘transformed data’ is needed
The explanation requested has been added.
line 137 – the explanation of applying Bonferroni correction is needed
More detail about the application of the Bonferroni correction has been added.
Additional comment regarding formatting of the paper:
The paper is delivered in almost unreadable way. The corrected parts should be marked by different colour, not by following changes mode.
The previous revisions have been removed.
Reviewer 2 Report
The authors clarify my concern left in the first round of review. In fact, considering that the manuscript is a short Communication, there would be no problem. However, and now that the authors have clarified the novelty, it would be good to make it completely clear in the Introduction and part of the Conclusions. I invite the authors to clarify this research line and that with every new publication the authors clarify the new developments reported with each new publication.
Author Response
As suggested, more explanation was added both in the “Introduction” and “Conclusions” sections. Moreover, few minimal typing mistakes were corrected. All changes were highlighted using violet color text.
Reviewer 3 Report
The authors have improved their manuscript. Still, it is not clear how different air-classified fractions F and G were obtained. Authors should clearly show it Figure 1 and/or 2. Also, in Figure 1, measuring unit is missing .. from 220 to 280...
Author Response
The authors have improved their manuscript. Still, it is not clear how different air-classified fractions F and G were obtained. Authors should clearly show it Figure 1 and/or 2. Also, in Figure 1, measuring unit is missing .. from 220 to 280...
With the aim to better clarify how the F and G fractions were collected, more explanation was added in the caption of Fig. 3 concerning the air-classifier image. Moreover, Fig. 1 was improved through the specification of the airflow setting from 220 to 280 values. All integrations were highlighted using violet color text.